# Institutional Lens upon Industrial Symbiosis Dynamics: The Case of Persian Gulf Mining and Metal Industries Special Economic Zone

**Shiva Noori \*, Gijsbert Korevaar and Andrea Ramirez Ramirez**

Department of Engineering Systems and Services, Faculty of Technology, Policy and Management, Delft University of Technology, 2628 BX Delft, The Netherlands; g.korevaar@tudelft.nl (G.K.); c.a.ramirezramirez@tudelft.nl (A.R.R.)

\* Correspondence: s.noori@tudelft.nl; Tel.: +31-633998578

**Abstract:** Industrial Symbiosis (IS) is a collaboration between nearby industrial plants to exchange waste material and energy and achieve economic and environmental benefits that cannot be obtained individually. IS emergence in a cluster requires both technical potentials for material and energy exchange and social readiness for collaboration. In this paper, to gain insight into IS dynamics in emerging industrial clusters; we investigate shared concepts governing actors' behavior in the form of rules and regulations, and social norms and practices. We implemented the IS dynamics framework to reveal which dynamics are supported either by the legislation or actors' preferences. The Persian Gulf Mining and Metal Industries Special Economic Zone in Iran is used as a case study. The case study revealed that previous successful collaborations in the cluster were often self-organized, but stakeholders preferred to initiate new IS collaborations if financial incentives and infrastructure are provided. Meanwhile, the institutional analysis showed that institutional arrangements (e.g., pricing and penalties) are not in favor of IS emergence. Even though stakeholders might engage in self-organized IS because of inherent problems such as resource scarcity, the lack of clear and effective institutions could hinder IS. This understanding can help both the government and stakeholders in their strategies for future collaborations under different economic and environmental policies.

**Keywords:** industrial symbiosis; dynamics; institutions; drivers

## 1. Introduction

Industrial systems are not only embedded in natural ecosystems [1], but are also entwined with human society, shaping social construction [2]. In Industrial Symbiosis (IS), this becomes more notable as stakeholders' behavior is affected by other stakeholders in the network, as well as social norms and practices [3]. Industrial Symbiosis is a collaborative relationship in which two or more nearby industrial plants exchange co-products, by-products, waste material, or waste energy to achieve economic and environmental benefits that cannot be obtained individually [4]. IS emergence is a phase of IS evolution in which stakeholders become aware of IS opportunities, explore new connections, and look for potential partners to build symbiotic relationships [5]. The successful IS emergence in a cluster needs both opportunities for material and energy exchange as well as opportunities for collaboration.

All IS practices have not emerged through the same evolution pathways. In this study, we call these evolution pathways dynamics. In general, IS dynamics could be categorized into self-organized, facilitated, and planned. Self-organized IS is initiated by involved stakeholders themselves, while planned IS is regulated by governmental or regional development policies. Facilitated IS is

coordinated and administered by a third party [4]. Boons et al. developed a more detailed approach to IS evolution pathways and recognized seven IS dynamics worldwide, characterized by initial stakeholders, their motivations, and events leading to IS contact. In this framework, IS initiated by industrial stakeholders could lead to self-organization or organizational boundary change dynamics. Facilitated IS passes through brokerage, collective learning, or pilot facilitation and dissemination dynamics. Finally, planned IS is classified into government planning and eco-cluster development dynamics [6]. Later, Sun et al. [7] defined anchoring activities to refer to the effort of local stakeholders creating favorable technical and institutional conditions for IS emergence. In this study, we considered anchoring as an eighth dynamics mainly to address physical and social facilitation in the cluster.

An Emerging Industrial Cluster (EIC) is a cluster in the first stages of development, while expected to expand rapidly [8]. EICs play a significant role in the industrialization of developing countries. Existence of technical potential for symbiotic exchange in an EIC does not result in a unique IS dynamic, since institutional and geographical conditions play an influential role in IS emergence [5]. Institutions are "shared concepts used by humans in repetitive situations" [9]. Laws and regulations are very important to promote or hinder industrial symbiosis emergence [4]. Recent studies have aimed to examine the role of institutions in IS emergence. In extensive research, to investigate the development of IS networks in seven existing IS examples, Mileva-Boshkoska et al. [10] have studied institutions within a multi-attribute decision-making model, along with non-social aspects. Fraccascia et al. have confirmed that policy measures such as economic subsidy and landfill tax support the emergence of self-organized IS. Some other researchers focused on the role of national institutional arrangements in IS development. For instance, the weakness or geographic variation of financial incentives, and legislative problems are also recognized as essential IS development barriers in Europe [11]. In the Finnish legislation system, there is a need for an innovative by-product assessment procedure [12] while the National Industrial Symbiosis Program (NISP) in the UK promotes IS through institutional capacity building [13]. However, as various interpretations of IS in different countries exist as a result of diversity in economic and legal systems [6], there is a need for a systematic institutional analysis in the IS field to understand how institutions could lead to different IS dynamics.

IS emerges in a cluster as a consequence of particular interactions at the network level such as orientation, feasible study, planning, and implementation [14]. These pre-emergence interactions create common ground among stakeholders [15] and improve relational, knowledge, and mobilization capacity of the cluster [14]. As the interaction proceeds, trust and shared vision are created between the organizations [16]. In a supplier–buyer relationship, interactions such as joint efforts and dedicated investments tighten the relationship between the stakeholders [17]. Intentional pre-emergence network development provides a platform for IS emergence [18]. Frequent interaction, tacit knowledge exchange, and shared culture among the organizations provide favorable conditions for IS emergence. [16] These studies emphasize the role of pre-emergence collaborations in IS, but it is not clear if such collaborations could promote a particular IS dynamics in the future development of EICs.

According to Beckert's social field theory, social networks, institutions, and cognitive frames are three social forces that shape socially embedded markets [19]. While many previous studies investigated existing IS networks, this research focuses on emerging industrial clusters (EICs) in which IS is not shaped yet. We adopt the IS dynamics framework by Boons et al. to investigate social forces shaping IS relationships in the form of pre-emergence collaborations, institutions guiding those collaborations, and capture stakeholders' motivation to engage in IS. Assuming that the technical potential for symbiotic exchange already exists in an EIC [20], we aim to understand which IS dynamic could be expected to emerge. For this purpose, previous collaborations in an EIC and the structure of successful ones are studied through a questionnaire to all stakeholders. Then, the same group of stakeholders are asked under which conditions they will engage in IS collaboration. In parallel, regional and national regulations governing industrial activities are investigated by systematically analyzing a selection of laws and regulations (e.g., environmental regulations and

energy prices). Finally, we examine which IS dynamic is more likely to emerge by comparing the outcomes of the previous three steps in the IS dynamics framework. We employ the Persian Gulf Mining and Metals Special Economic Zone, Iran as a case study of an emerging industrial cluster. The paper holds six sections. Section 2 explains the theoretical background, Section 3 briefly introduces the case study, Section 4 describes the methods, and Section 5 presents the results. Section 6 states the contribution of this work to IS social studies.

## 2. Theoretical Background

### 2.1. IS Drivers

Even though the availability of matching waste sources and sinks is the starting point for symbiotic exchanges, different drivers can promote IS in various regulatory and institutional contexts [21]. To study IS, it is crucial to understand which motives and mechanisms influence IS emergence [14,22]. IS drivers have been studied extensively in the literature. (e.g., [22–26]). IS dynamics framework [6] also considers actors' motivation determinant in IS dynamics as indicated in the second column of Table 1. To implement the IS dynamics framework in practice, we made an inventory of IS drivers from the literature and checked each driver matches which dynamic's motive. For instance, 'Information about IS opportunities and potential synergy partners' [22] matched market transparency and 'Workshops, conferences, seminars, and forums' [26] suited knowledge development. Table 1 shows how IS driver literature is linked to the IS dynamics framework.

**Table 1.** Industrial Symbiosis (IS) drivers linked to IS dynamics—author generated.

| IS Dynamics [6,7] | Motive [6,7] | Drivers from literature [21–26] |
|---|---|---|
| Self-organization | Economic/ environmental benefit | • Energy and utility supply costs/Waste disposal cost <br> • Redundancy in energy, water, and material supply <br> • Resource scarcity <br> • Increasing eco-efficiency of the company |
| Organizational boundary change | Business integration/separation | • New business opportunities |
| Facilitation—brokerage | Market transparency | • Feasible studies <br> • Information about IS opportunities and potential synergy partners |
| Facilitation—collective learning | Knowledge development | • Community awareness about environmental and economic impacts of the companies <br> • Workshops, conferences, seminars and forums |
| Pilot facilitation and dissemination | Best practice development by piloting | • Other successful IS experiences in the cluster <br> • Learning from non-local IS experiences |
| Government planning | Governmental control and command | • Governmental plans for IS implementation <br> • Stimulation policies/incentives/subsidies by the government <br> • Monitoring and environmental assessment by governmental organizations |
| Eco-cluster development | Regional economic development | • Regional policies to transform the cluster into EIP <br> • Innovative solutions for cluster development |
| Anchoring | Physical and social anchoring | • Infrastructure readiness <br> • Managerial support <br> • Social interactions |

*2.2. ADICO; the Grammar of Institutions*

A socio-technical system is a dynamic entity composed of stakeholders and technical artifacts interacting in interdependent physical and social networks [27,28]. Stakeholders' perceptions and interactions in a socio-technical system are guided by institutions [29]. Institutions are expressed in the form of rules, norms, and strategic visions in human behavior patterns. These linguistic expressions are called Institutional statements. "Institutional statement refers to shared linguistic constraint or opportunity that prescribes, permits or advises actions or outcomes for stakeholders" [30].

Crawford and Ostrom [30] introduced a grammatical syntax to analyze institutional statements called ADICO. ADICO refers to five components of institutional statements, which are attribute (A), deontic (D), aim (I), condition (C), and or else (O). Different combinations of ADICO syntax shape three types of institutional statements: Rules include all components (ADICO); norms consist of the attribute, deontic, aim, and condition (ADIC); and shared strategies only have the attribute, aim, and condition (AIC) [30]. In the IS dynamics framework, each dynamic is characterized by its initial stakeholder and overall storyline [6]. ADICO grammar provides a lens to study institutional statements in terms of attribute and aim and link them to the stakeholder and storyline in the dynamics. Using ADICO, legal enforcement for IS implementation could be examined by studying deontic and sanctions. Often, institutions are not expressed in separate institutional statements but nested within other institutions. Basurto et al. [31] set practical guidelines explaining how to identify institutional statements in legislation systematically, code and classify institutional statements using ADICO grammar, and analyze the coded data independently and nested. We implemented their method to discover and analyze institutional statements that form the basis for IS and reveal which dynamics are supported by the institutions.

## 3. The Case Study

The Persian Gulf Mining and Metal Industries Special Economic Zone (PGSEZ) is located in the southern part of Iran, 14 kilometers west of Bandar Abbas, at the Persian Gulf. The cabinet approved cluster establishment in this area in 1998. PGSEZ is a subsidiary of the Iranian Mines and Mining Industries Development and Renovation Organization (IMIDRO), which is a state-owned corporation itself. The cluster management provides common infrastructures, coordinates relationships, and supervises development plans of the established companies. PGSEZ is planned to be a hub of energy-intensive industries in the Middle East because of proximity to the South Pars, which is one of the largest natural gas reservoirs in the world [32]. Key established industries in the cluster, their main product, capacity, commissioning year, and principal shareholder are listed in Table 2. Main shareholders of PGSEZ management (PGS), South Kaveh Steel Company (SKS), Persian Gulf Saba (SAB), and Hormoz Power Plant (HPP) are state-owned organizations themselves. This composition of semi-governmental and private energy-intensive and polluting industries makes this EIC a suitable case for IS implementation as a strategy for sustainable development. Almost all companies have expansion projects in construction or feasibility study phase. Furthermore, new companies are also planned to be established in this cluster. However, industrial development in Iran struggles with US sanctions on metal industries, water scarcity [33,34], and high $CO_2$ emissions [35].

**Table 2.** Located organizations in the Persian Gulf Mining and Metal Industries Special Economic Zone (PGSEZ), their capacity, ownership, and establishment year (http://www.pgsez.ir).

| Organization | Main product | Capacity | Start Year | Main Shareholder |
|---|---|---|---|---|
| PGSEZ management (PGS) | --- | --- | 1998 | IMIDRO |
| Hormozgan Steel Company (HOS) | Steel slab | 1,500,000 tonns/year | 2009 | Mobarakeh Steel Company |
| South Kaveh Steel Company (SKS) | Steel billet | 1,200,000 tonns/year | 2012 | Mostazafan Foundation of Islamic Revolution |

| Persian Gulf Saba (SAB) | Direct reduced iron | 1,000,000 tonns/year | 2017 | Civil Pension Fund Investment Company |
|---|---|---|---|---|
| Maad Koosh Pelletizing Plant (MKP) | Iron pellet | 2,500,000 tonns/year | 2018 | Arzesh Holding |
| Almahdi Aluminum and Hormozal Complex (AAC) | Aluminum ingot | 172,000 tonns/year | 1990 | Mapna group |
| Hormoz Power Plant (HPP) | Electricity | 160 MW | 2018 | Ghadir Investment Company |

## 4. Methods

In socio-technical analysis, stakeholders could be individuals or organizations [4]. In this work, we studied organizational stakeholders. To get insight into the IS emergence dynamic in the cluster, we investigated previous collaborations between stakeholders, their motivations to start new IS collaboration, and institutions governing their activities. This study was conducted in two phases. In the introductory study, we obtained a general overview of the current structure of the cluster and its environmental problems via site visits and open interviews. Then, data for the in-depth study were gathered via questionnaire and desk research. Through the questionnaire, we aimed to understand the dynamics of previous collaborations in the cluster. IS drivers were also studied to figure out the preferred dynamics for future IS, based on Table 1. In desk research, we investigated the institutional context of the cluster in the form of industrial energy and water prices, national regulations governing energy, and environmental-related issues in the industry. We analyzed the institutions using ADICO grammar to understand which IS dynamic is supported by official regulations. Finally, the field observations, survey, and desk research outcomes were compared and criticized from the viewpoint of the dynamics to get insight into IS dynamics in the cluster. We investigated mismatches to reveal IS emergence barriers. A detailed description of the methods is given in Sections 4.1–4.3.

### 4.1. Introductory Study

During the introductory study carried out in summer 2018 in Iran, we interviewed experts from industries, cluster management, consultation companies, and representatives from governmental policy-making organizations at the industrial site of PGSEZ. To maintain flexibility, we conducted open personal interviews to understand how current collaborations among stakeholders have been shaped and reveal drivers and limitations for sustainable development of the cluster. Each interview took around 45 to 60 min. The interviews were conducted in Persian. While gathering technical data about gaseous, liquid, and solid wastes, we also looked into the monitoring, treatment, and disposal of such pollutants in the field. Notes taken during interviews and site visits were summarized and translated to English in field observation report [36]. This introductory study helped us to get insight into the structure, power hierarchy, collaborations, and barriers to sustainable development in the cluster. It also revealed some shared strategies especially about the role of cluster management and province-level governmental organizations in cluster development.

### 4.2. Focused Survey

4.2.1. Survey Design

To study previous collaborations and IS drivers, a questionnaire was designed (See Appendix A). The questionnaire covered four sections: General information, collaboration matrix, successful collaborations, and drivers of Industrial Symbiosis. Under general information, we asked about the respondent's organization, occupation, and experience related to the current job to assure they have sufficient knowledge and expertise to participate in the survey. In collaboration, two or more stakeholders share tangible (e.g., money, physical asset) or intangible resources (e.g., insights, knowledge, and authority) to solve problem or attain benefits greater than working isolated [17]. Based on Boons et al. [14] and Nyaga et al. [17], we defined three categories of collaboration in the questionnaire: (a) Knowledge sharing (including technical advice, supervision and project

management), (b) trade, (including main and by-product trade), and (c) dedicated investment (covering utility supply, and joint investment). The collaboration matrix aimed to capture different types of collaboration between every two organizations in the cluster during the last five years. In the third part, we asked the respondents to consider one of the most successful collaborations their organization has in this period and indicate the contract type, involved organizations, initiator, facilitator, and communication method of that collaboration. The quality of such collaboration was also reviewed in terms of its influence on shared strategic vision, long-term relationships, and information exchange among organizations. Finally, based on Table 1, the respondents were asked to indicate which parameter would encourage their organization to start new Industrial Symbiosis collaborations with existing or future companies in the PGSEZ. This part was designed in a five-point Likert scale. The questionnaire was translated to Persian. For clarity, the concepts of collaboration and IS were defined at the beginning of the questionnaire.

### 4.2.2. Sampling and Data Collection

We collected the research data from surveys distributed among managers of the business and non-business organizations in the cluster in November and December 2019. Before sending the questionnaire to the main respondents, we tested it by distributing among a sample group in academia, which was familiar with the IS concept. The potential respondents were selected among the company, plant operation, energy, infrastructure, and technical managers in each organization to ensure all types of collaborations were reported. The managers each had more than three years of experience in the cluster. First, an invitation email, explaining the purpose of the survey, was sent to the suggested respondents to ask if they participate in the survey. In the case of negative or no response, we looked for other potential respondents. After ensuring at least three managers from three different departments of the organization were willing to fill the questionnaire, the link to the online survey was sent to them.

### 4.2.3. Data Analysis

The collected data were analyzed qualitatively and quantitatively. After collecting the filled questionnaires, we checked the occupation and experience of the respondents in their current occupation, which ensured us they have enough knowledge and expertise about previous collaborations in the cluster. Based on the collaboration matrix, we mapped the collaborations network as a graph. For collaborations between two organizations, we combined reported collaborations by the respondents from both organizations. This representation enabled us to examine which stakeholders have mostly collaborated, and which types of collaborations were prevalent in the cluster. Stakeholders who have historically bridging roles are also more likely to start new collaborations [15]. We checked whether such stakeholders exist in the cluster, especially looked at the cluster management. We also examined if the structure of the graph was more homogenous or preferential. In planned IS, network growth is mostly homogenous while it is preferential in self-organized IS [37].

Analyzing the third part of the questionnaire, we especially aimed to trace the role of cluster management and the government in initiating, facilitating, and monitoring previous successful collaborations. We also investigated the formality of the collaborations in its contract and information exchange method. This part helped us to understand whether previous collaborations were often facilitated, planned, or self-organized. We also evaluated if shared vision, long-term relationships, and communication among the organizations have improved or damaged as a result of the collaboration. As Stated in Section 1, improvement in those aspects provides a platform for IS emergence in the cluster. The last part of the questionnaire was designed to reveal under which circumstances the organizations would engage in IS collaboration. In Likert scale, "very" and "completely" were considered to highly motivate IS, while "not at all" and "slightly" supposed to have a low impact on IS emergence. According to Table 1, we matched the driver with IS dynamics to see which dynamics are more preferred by the organizations. Comparing results of part three and four,

we checked if the governing dynamics of previous collaborations matched with stakeholders' preferences for future IS relationship or not.

*4.3. Desk Research*

Complementary desk research was conducted to gain insight into institutions governing industrial activities and cluster development. For examining environmental institutions, we checked the collection of environmental laws and regulations. The collection has two parts: National laws and international conventions and treaties. In this paper, we focused on national laws. This part covered hunting, farming, tourism, trade, urbanization, and industrial activities. We limited the scope to the regulations related to the industry sector and ignored those related to a specific province. Finally, we ended in 10 documents. In the energy sector, we reviewed the "energy consumption pattern reform law". The scope of this law is management and optimization of all produced, imported, or consumed energies in the country to improve efficiencies, avoid losses, protect the environment, and support sustainable development. We also studied the executive procedure of clause 26 of this law, which defines penalties for disobedient industries. To industrial development, we investigated the rules and regulations for the establishment of production, industrial, and mining units. The list of 13 documents was sent to a few experts in the cluster to ensure all laws and procedures are currently active.

For ADICO coding, all clauses, paragraphs, and notes were copied to an excel sheet indicating the title of legislation, its issue or approval date, clause number, and note number. The list was generated in Farsi, and ADICO components were extracted from Farsi statements. Since headings, introductions, and definitions do not form institutional statements, we ignored them while tracing the legislations. In case a statement had several parts, we decomposed it to a few statements [31]. Then, we checked if the institutional statement is relevant to IS or not. Our criteria was relevance to the industry section, industrial clusters, waste management, and energy consumption and recovery in the industry. In total, 183 IS-related statement were recognized. Then, attribute, deontic, aim, condition, and sanction of those statements were identified. Many statements were passive. We decided about the attribute according to the other sentences in the legislation. In the case of ambiguity, all possible attributes were listed. We also indicated whether the statement is a rule, norm, or strategic vision. Within a broad legal system, which allows sanctions for disobeying, all written legislation might be regarded as rules. In this paper, however, we studied institutional statements as "nested elements that operate at one level as a whole system and at a different level as part of another complex system". To identify norms from strategies, we accepted implicit deontic. Thus, when deontic was not explicit in the statement but linked to previously stated deontic in the legislation, we considered the latter one as a norm. For instance, the first clause of the "Air pollution prevention law" and "Waste management law" obliges all organizations and individuals to follow regulations and policies described in these laws. Thus, we considered this obligation for all clauses in the law. When the aim of the statement was "subjected to" or "dependent on" another activity, this was assumed as an obligation, therefore categorized as a norm. The terms "in charge of" and "responsible" were also assumed as obligation for the stakeholders. We divided norms into obligation, permission, and prohibition. Then, we checked when and where the institutional statement is applicable. Finally, the penalties and sanctions in case the institutional statement is not followed were checked.

Table 3 shows how ADICO-coded institutional statements were analyzed in line with IS dynamics framework. In attribute, we checked if a statement refers to the industries, cluster management, or the government and governmental organizations. As the aims of the institutional statements were too diverse, we categorized them in a few topics [31] and linked them to IS dynamics. These topics were 'Pricing' (e.g., pricing of energy, material, waste disposal), 'Eco-efficiency improvement' (e.g., technical improvement, environmental protection, and energy efficiency), 'Infrastructure provision', 'Market brokerage', 'Knowledge development and awareness' (e.g., training, information sharing), 'Economic stimulation' (e.g., tax cut, loan), 'Industrial and cluster development' (e.g., distance from cities, industry classification), 'Regulatory and legislation' (e.g., defining new standards and execution procedures), and 'Environmental monitoring and assessment'

(e.g., effluent measurement, self-declaration). Institutional statements aimed at pricing and eco-efficiency improvement were regarded to support the self-organized dynamics. Statements with economic stimulation, legislation, and environmental monitoring and assessment topics were considered to promote government planning dynamics. Market brokerage, knowledge development, industrial and cluster development, and infrastructure provision topics were related to facilitation-brokerage, facilitation-collective learning, eco-cluster development, and anchoring dynamics respectively.

**Table 3.** ADICO grammar linked to IS dynamics—author generated.

| Attribute | Deontic | Topic | Condition | Or else |
|---|---|---|---|---|
| Government/Governmental organizations | Obligation | Pricing | When | Penalties |
| | Permission | Eco-efficiency improvement | Where | Sanctions |
| Cluster management | Prohibition | Infrastructure provision | If | |
| Industries | | Market brokerage | Unless | |
| | | Knowledge development and awareness | | |
| | | Economic stimulation | | |
| | | Industrial and Cluster development | | |
| | | Regulatory and legislation | | |
| | | Environmental monitoring and assessment | | |

After this classification, we analyzed the legislations in terms of attribute and their topic to examine whether each statement supports any IS dynamics. We looked especially for the role of governmental organizations and cluster management to facilitate or enforce material and energy exchange between the industries. The penalties and sanctions in case the rules are not followed were also investigated in depth separately. When it was nested in another institution, that institution was also checked to clarify the sanctioning. Discretionary imprisonments were converted to equivalent fines, and the fines calculated in Euro. Since many sanctions were determined as a percentage of prices, we considered prices also as institutions and gathered industrial energy and water prices during the last five years (2015–2019). Any ambiguities in the statements and conflicts in the in the sanctions were also investigated.

## 5. Results

The questionnaire was distributed among 21 managers in the cluster and received 13 filled questionnaires back from six out of seven active organizations. No one from AAC filled the questionnaire. The majority of the respondents were from management, and development and planning departments. There were also respondents from operation, energy and utility, engineering, and environmental protection departments. They had, on average, around six years of experience in their position. The position and experience of the respondents ensured us that they have sufficient knowledge about the topic of the questionnaire.

### 5.1. Previous Collaborations

Previous collaborations in the cluster are visualized in Figure 1 as a multigraph. Three categories of collaboration, knowledge sharing, trade, and dedicated investment, are indicated by letters from a to c respectively. For AAC, we used the reported collaborations from the other companies to map the graph. Almost all types of collaborations have taken place in the cluster during the last five years. Collaborations were mainly shaped between the cluster management and three steel industries: HOS,

SKS, and SAB. The three other organizations (HPP, AAC, and MKP) have rarely collaborated with other organizations in the cluster. This shows that the network structure was preferential rather than homogenous, which resembles a self-organized dynamic.

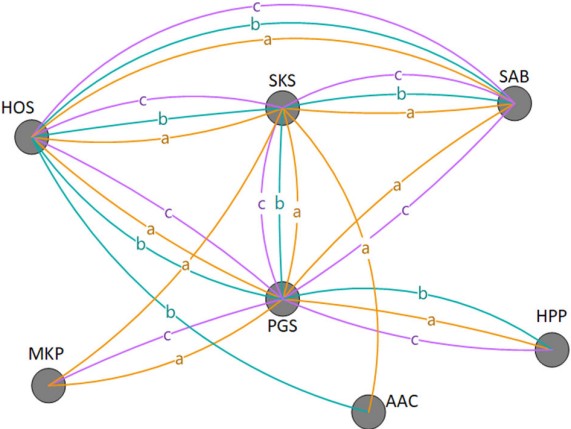

**Figure 1.** Previous collaborations network in PGSEZ: (a: Knowledge sharing; b: Trade; c: Dedicated investment).

AAC is the first company established in the area, before the area was transformed officially into an industrial cluster, but it has had only two collaborations with HOS and SKS during the last five years. Hormoz Power Plant (HPP) had the largest technical potential for energy exchange [20], but it has been reported to have no collaboration with the other industries in the cluster during the last five years of construction and operation. On the other hand, SAB is almost new in the cluster but has significantly collaborated with two other steel industries. The number and diversity of collaborations for HOS and SKS showed that these stakeholders are likely to engage in further collaborations.

Several technical advice and consultation interactions in the cluster represent capability for knowledge transfer, which is essential for IS emergence. By-product trade has not occurred frequently between the companies, which could be because of limited technical potential for it [20]. All stakeholders, unless ACC, had an investment collaboration with the cluster management since it is responsible for utility supply to the industries. However, this category of collaborations was not limited to cluster management. SKS, SAB, and HOS collaborated in the form of utility supply or joint investment, which shows their tendency to tangible resource sharing.

Figure 2 presents the structure of previous successful collaborations in terms of other involved organizations, initiator, and infrastructure provider. Province-level governmental organizations were involved in successful collaborations more than the other listed organizations. Despite Figure 1 showing cluster management collaborated with the other organizations, it was reported to be involved only in around 30% of successful collaborations. The organizations themselves started these collaborations and provided the required infrastructure for it. It could be said that previous collaborations were more likely to be self-organized. We also observed formality of previous collaborations in terms of contract type and communication method. Formal agreements were routine in successful collaborations. The preferred way of communication was formal meetings over other means (e.g., email, phone call, and social media). The most successful collaborations in the cluster were long-term with an average length of six years, mostly still ongoing. The respondents collectively answered that these collaborations have created or developed shared strategic vision, long-term relationship, and information exchange base between the organizations which have proven positive influence on IS emergence. Diversity of previous collaborations, duration of successful ones, and relational improvement because of such collaborations show that a basis for IS emergence has been created in the cluster.

## 5.2. Drivers of Industrial Symbiosis

In the questionnaire, increasing material productivity or energy efficiency and decreasing Greenhouse Gas (GHG) emissions were regarded as environmental benefit drivers, which were found to moderately drive stakeholders to initiate IS collaboration. An increase in resource costs or waste disposal costs was not also a strong driver for IS collaborations. Among self-organized IS drivers, only resource scarcity was considered very likely to motivate the organizations. This was compatible with field observation, which showed resource scarcity played an influential role in current collaborations in the cluster. For instance, land shortage near the sea for desalination units has caused water trade between the industries.

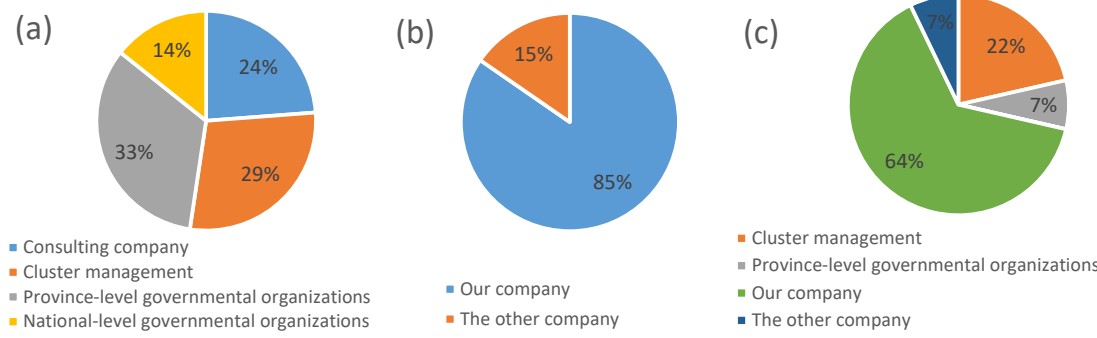

**Figure 2.** Previous successful collaborations in the cluster: (**a**) The other involved organizations; (**b**) initiator; (**c**) infrastructure provider.

Most managers pointed out that infrastructure readiness and governmental financial stimulation policies (e.g., subsidies or tax cuts) were very likely to promote IS in their organization. Thus, anchoring and government planning dynamics were the most preferred dynamics by the stakeholders for future IS collaborations in the cluster. The next influential drivers were information about available waste heat and material in the cluster for exchange, information about other successful industrial symbiosis projects, and cluster development plans organized by the cluster management. It shows that IS might also emerge in the cluster as a result of market transparency, pilot facilitation, or eco-cluster development plans. Monitoring and environmental assessment by governmental organizations and short or long-term business opportunities were almost a moderate motivation for organizations to participate in new IS connections. Surprisingly, an increase in resource prices or waste disposal cost was not among the dominant drivers of IS emergence. To understand the reason, we investigated these two parameters in more detail in institutional analysis. Collective learning was the lease favored dynamic by the stakeholders. Therefore, the most preferred dynamics by the stakeholders are the ones that are supported by the government. However, the cluster management can also facilitate IS through market brokerage, sustainable development plans, and introducing successful IS experiences to the cluster.

Coming to each company, the drivers were slightly different. For instance, supportive policies by the government and cluster management were found less important for HOS, but new business opportunities were more significant for this company. SKS managers paid higher attention to economic benefit than other respondents. This shows that HOS and SKS are more interested in industry-initiated dynamics. The number and diversity of collaborations of these two companies during the last five years also reflected this approach. The cluster management had a minor concern about economic benefit but showed a high interest in eco-efficiency improvement. They mentioned that environmental monitoring by the government considerably motivates IS. The other companies followed more or less similar motives of the whole cluster.

### 5.3. Institutional Analysis

In Table 4, we have listed investigated laws and regulations, indicating the clauses governing industrial activities, the total number of IS-related statements, and number of rules, norms, and strategies. A sample of ADICO coded statements translated to English is available in Appendix B. With assumptions in Section 4.3, 19 out of 183 statements were identified as rules, 137 as norms without clear sanctioning in case of disobedience, and 27 as strategies. From 137 norms, 114 were obligatory, 15 permissive, and 8 prohibitive.

**Table 4.** List of investigated regulations and number of IS-related rules, norms, and strategies [38].

| | Legislation | Issue/Approval Year | Applicable to the Industries | IS-Related Statements | Rules | Norms | Strategies |
|---|---|---|---|---|---|---|---|
| 1 | Fifth country development plan | 2011 | Clause 192 | 6 | 1 | 5 | 0 |
| 2 | Sixth country development plan | 2017 | Clause 35 to 50 | 19 | 0 | 18 | 1 |
| 3 | Air pollution prevention law | 1995 | Chapter 3 | 24 | 1 | 23 | 0 |
| 4 | The executive procedure of air pollution prevention | 2000 | All | 0 | 0 | 0 | 0 |
| 5 | The executive procedure of environmental impact assessment of large manufacturing, service, and development plans and projects | 2011 | All | 6 | 2 | 4 | 0 |
| 6 | The executive procedure of water pollution prevention | 1994 | All | 19 | 1 | 12 | 6 |
| 7 | Waste management law | 2004 | All | 13 | 4 | 9 | 0 |
| 8 | The executive procedure of waste management | 2005 | Clause 12, 28, 30, 31, 32 | 10 | 2 | 6 | 2 |
| 9 | Value Added Tax (VAT) law | 2008 | Clause 38 | 3 | 1 | 1 | 1 |
| 10 | Soil Protection law | 2019 | Clause 13 | 12 | 4 | 7 | 1 |
| 11 | Energy consumption pattern reform law | 2011 | All | 33 | 1 | 25 | 7 |
| 12 | Executive procedure of clause 26 of energy consumption pattern reform | 2014 | All | 10 | 1 | 7 | 2 |
| 13 | Rules and regulations for the establishment of production, industrial and mining units | 2018 | All | 28 | 1 | 20 | 7 |
| | **Total number** | | | **183** | **19** | **137** | **27** |

The institutions were also classified by attribute and topic and consequently linked to dynamics as visualized in Figure 3. Coming to the attribute, 92 statements referred to the government or governmental organizations, 82 belonged to the industries, and only 5 addressed the cluster management directly. Four statements were also recognized to apply to all actors. Among governmental organizations, the DOE and ministries, especially the ministries of energy and petroleum, were the focal attributes.

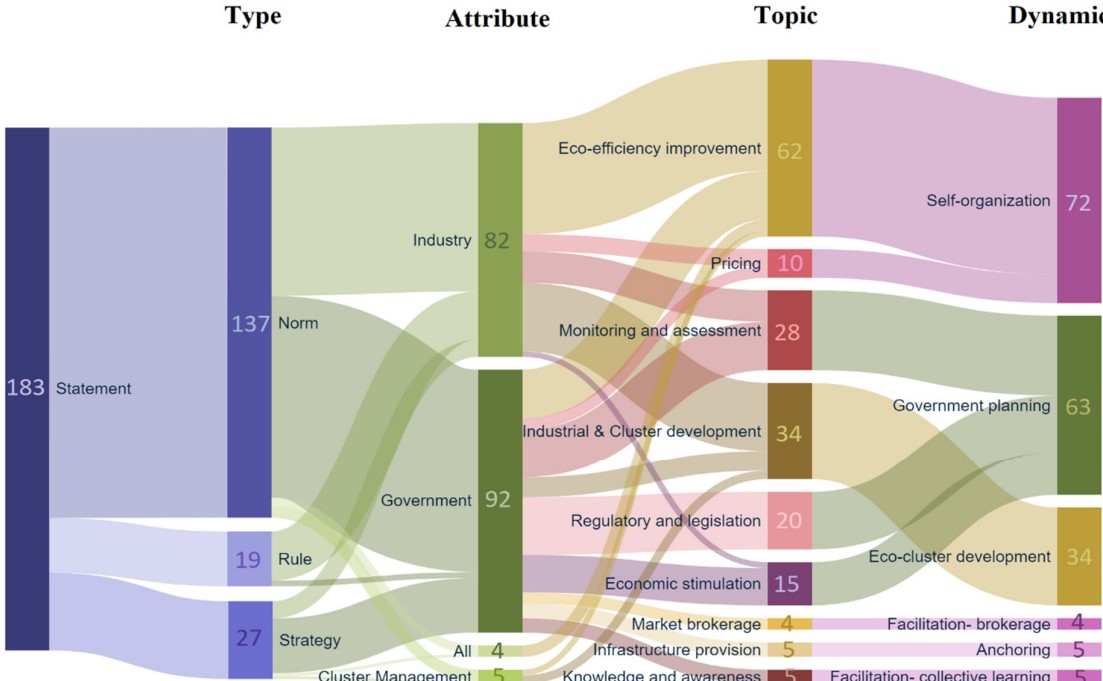

**Figure 3.** Distribution of institutional statements by type, attribute, and dynamics.

IS was not addressed in the legislation directly. We considered environmental monitoring and assessment, economic stimulation, and regulatory and legislation to support governmental planning dynamic. Regarding the total number of statements, this dynamic was supported by institutions. Even though as per written institutions, monitoring air, water, and soil effluents, and notifying lawbreaker industries is DOE's responsibility, we realized in the field observation that the provincial environmental department does not monitor water and air effluents adequately. In such a condition, the questionnaire also revealed that the organizations do not seem to consider environmental monitoring a powerful driver for IS. Technical improvement in the form of energy efficiency improvement and material productivity improvement had been obliged to both the industries and the government. The institutional study showed that even though the country has joined the Paris Agreement, there is not effective regulation about industrial GHG emissions in legislation. That might explain why, as per the questionnaire outcomes, decreasing GHG emission is not currently a driver for material and energy exchange among industries. Furthermore, collective learning dynamic was rarely addressed in the legislation. Stakeholders were not interested in this dynamic for future IS collaborations. We did not find any institutional statements governing pilot facilitation and dissemination, and organizational boundary change dynamics.

In market brokerage topic, only few regulations were found aiming electricity trade between the industries and the ministry of energy. This IS drivers study showed market transparency and information about available waste energy and material for recovery, which can encourage industries to initiate IS. Anchoring via infrastructure provision was the favored dynamic for future IS in the cluster, but this dynamic was not supported strongly in the official institutions. Infrastructure provision in the form of desalination units, wastewater treatment units, and electricity for the industries was in the scope of government and ministry of energy, but previous successful collaborations showed that cluster management or governmental organizations did not afford infrastructures. Even though eco-efficiency improvement was seen to be highly supported by the institutions, the stakeholders appeared not interested in it as a driver for IS collaboration (Section 5.2). Lack of legal enforcement on environmental regulations found out in the field observation, low energy prices, and negligible penalties for environmental effluents can describe the insignificant concern about eco-efficiency improvement.

As discussed in Section 5.2, the respondent managers indicated that financial stimulation policies by the government (e.g., subsidies or tax cuts) significantly promote their organizations toward IS. However, we found only 15 statements aiming at economic stimulation in the institutions. More investigation showed that incentives are often vague and undetermined in the legislation. Although the government, Ministry of Energy, Ministry of Petroleum, and the Energy Council have to provide economic incentives for the industries to improve their energy efficiency and install energy recovery systems, such incentives are not clearly defined. Industrial development in the form of clustering was highly recommended in the regulations. The technical study [20] showed that cluster management could improve the technical potential for IS by introducing new industries to the cluster. As stakeholders also admitted cluster development plans by cluster management, eco-cluster development was recognized as a probable IS emergence dynamic in this cluster, supported technically as well as by institutions and stakeholders.

Then, the sanctions were investigated in more detail. Despite the high inflation rate in Iran [39], most fines remained fixed in the legislation. For instance, the maximum fine for air pollution was only around 35 Euro, soil contamination 70 Euro, and waste disposal 690 Euro. According to VAT law, industries that do not follow environmental protection standards and regulations have to pay one percent of their sales price as pollution tax too. The penalty of energy consumption exceeding national standards was a percentage of energy prices. We checked industrial electricity, natural gas, and water prices in Hormozgan province in 2015 to 2019 [40]. Euro to Rial exchange rate had a steep increase during the last two years, but energy prices have not increased proportionally. Thus, we recognized a decreasing trend in the prices. Furthermore, energy costs were significantly lower than in developed economies. (e.g., the electricity price was one-tenth of the Netherlands [41] and natural gas price one-fifteenth [42]). As stated in Section 5.2, the rise in energy supply or waste disposal cost was not supposed by the stakeholders to boost IS in the cluster. This can explain why an increase in such costs cannot encourage industries to exchange waste energy and material.

## 6. Conclusions

This paper aimed to gain insight into IS emergence in EICs through IS dynamics framework. It investigated previous collaborations in the cluster, stakeholders' motivations to initiate new collaborations, and institutions governing industrial activities it the case of 'Persian Gulf Mining and Metal Industries Special Economic Zone' in Iran.

Pre-emergence conditions are acknowledged in the literature to affect IS emergence. The need for an integrated approach in studying pre-emergence conditions in EICs, where symbiotic exchanges have not shaped yet, led to this research. Study of previous collaborations in the case study revealed that existence of cluster management as coordination body in the cluster, does not guarantee facilitated collaboration between the companies. The stakeholder initiated successful collaborations and provided their required infrastructures by themselves. Long-term ongoing collaborations are considerable signs for future long-term relationships in the cluster, especially mentioning the point that such collaborations created or improved shared strategic vision and information exchange platform between the organizations.

A comparison of stakeholders' drivers and institutional analysis results showed a discrepancy between stakeholders' preferred dynamic for future IS collaboration and the supported dynamics by the regulations. Financial stimulation and infrastructure provision highly motivate stakeholders for IS, but institutional statements are unclear and ineffective in this regard. The institutions do not support market brokerage for symbiotic exchange properly. However, sustainable industrial development through clustering is highly recommended in the institutions and supported by the stakeholders. It could be said that although inherent challenges such as resource scarcity can promote self-organized IS in the cluster, in the absence of adequate economic and environmental institutions, stakeholders will not engage in IS collaboration. For sustainable industrialization, environmental rules and regulations must be improved continuously ahead of industrial growth.

However, this framework has also limitations. Looking for global IS evolution patterns, the framework focuses on network and institution level motivations and does not consider stakeholder

level drivers such as short mental distance [3], willingness to collaborate [26], and trust [25,43,44]. Data gathering for social studies was much more challenging than gathering technical data. Technical data might be available mid-level managers and engineering and operation staff, but collaboration data should be gathered from top managers of the organizations, which also had limited time and accessibility. As EICs are in the first stages of development, one apparent restriction in EIC research is the limited number of surveyed organizations and stakeholders. Recognizing the importance of empirical data in this work, we carefully selected the interviewees and respondents from involved stakeholders in previous and future collaborations.

Having the technical potential, stakeholders' readiness for collaboration, and institutions governing IS, the next step could be modeling the socio-technical structure of the cluster and explore alternative futures for IS that might occur under different conditions. This understanding guides regional and national industrial development policies toward more sustainable scenarios (Doménech and Davies, 2011). It also helps both companies and cluster management in their strategies for future collaborations in the cluster under different economic and environmental policies.

**Author Contributions:** Conceptualization, S.N., G.K., and A.R.R.; methodology, S.N., G.K., and A.R.R.; validation, G.K. and A.R.R.; formal analysis, S.N., G.K., and A.R.R.; investigation, G.K. and A.R.R.; resources, S.N.; data curation, S.N.; writing—original draft preparation, S.N.; writing—review and editing, G.K. and A.R.R.; visualization, S.N.; supervision, G.K. and A.R.R.; project administration, A.R.R.; All authors have read and agreed to the published version of the manuscript.

**Funding:** This research received no external funding.

**Acknowledgments:** We would like to thank Amirkabir University of Technology (AUT) and managing director of PGZEZ for their support on facilitating data gathering in the cluster.

**Conflicts of Interest:** The authors declare no conflict of interest.

## Appendix A

*The Questionnaire: Survey Regarding Collaboration Efforts among Companies in PGSEZ*

Introduction: Industrial Symbiosis (IS) is defined as collaborative relationship in which two or more nearby industrial plants exchange co-products, by-products, waste material or waste energy to achieve economic or environmental benefits that cannot be obtained individually. It can increase material productivity and energy efficiency and improve the corporate image of the whole cluster.

An Emerging Industrial Cluster (EIC) is an industrial cluster in its first stages of evolution which has unrealized possibilities for rapid growth. EICs play an influential role in the industrialization of emerging economies. This questionnaire is part of the Ph.D. research of Shiva Noori at Delft University of Technology, The Netherlands, on the evaluation of Industrial Symbiosis in emerging industrial clusters. The Persian Gulf Mines and Metals Special Economic Zone (PGSEZ) cluster has been selected as the case study in this research. Successful Industrial Symbiosis needs technical potential for material and energy exchange as well as social readiness for collaboration. In the first part of this PhD project, the technical potential for Industrial Symbiosis was investigated. In this part, we aim to assess cluster readiness for IS emergence. In this questionnaire, previous collaborations in the cluster and the enablers of successful ones will be studied first. Then, we would like to identify under which conditions actors will engage in IS collaboration. In parallel to this survey, we are investigating regional and national regulations governing industrial activities (e.g., environmental regulations and energy prices). The combination of the survey outcomes with desk research (literature review) will help us to understand the cluster readiness for IS emergence. This knowledge will help both companies and cluster management in their strategies for future collaborations in the cluster under different economic and environmental policies.

The survey results will be anonymized and all names and positions will remain confidential according to EU General Data Protection Regulation (GDPR). Data gathered through this survey will be analyzed together with institutional statements, and the results will be published in an academic journal. If you are interested, a summary of the results will be sent to you. For any complementary

data or clarification, you can contact me via contact information provided at the end of the questionnaire. Completing the survey will take about 15 minutes. It would be appreciated to receive filled questionnaire within a week.

Your name …………………………………………………………..

**Table A1.** General Information.

1    Please specify the company for which you work:

□The cluster management (PGS)          □Hormozgan Steel Complex (HOS)

□South Kaveh Steel Complex (SKS)        □Persian Gulf Saba Steel (SAB)

□Almahdi Aluminum company (AAC)     □Maad Koosh iron ore pelletizing company (MKP)

□Hormoz Power Plant (HPP)

2    Please specify the division in which you work:

□Management       □Engineering       □Energy & Utility      □ Development Planning

□HSE              □Operation         □ Others, please specify …………………………….

3    How long have you worked in this company?

Collaboration refers to the joint effort of two actors to share resources such as experience, knowledge, money, or physical assets to solve a problem or gain an advantage collectively. In this survey, we focus on intra-organizational collaborations. We have listed six collaboration types in the table below (Table A2. For each of the other companies in the PGSEZ cluster, please indicate in which way(s), if at all, your company has collaborated with them within the last 5 years.

**Table A2.** Collaboration Matrix.

| Collaboration with Company | Technical Advice & Consultation | Supervision and Project Management | Product Trade | By-Product Trade | Utility Supply(Electricity, Water, or Natural Gas) | Joint Investment | Other |
|---|---|---|---|---|---|---|---|
| PGS | □ | □ | □ | □ | □ | □ | □ |
| HOS | □ | □ | □ | □ | □ | □ | □ |
| SKS | □ | □ | □ | □ | □ | □ | □ |
| SAB | □ | □ | □ | □ | □ | □ | □ |
| AAC | □ | □ | □ | □ | □ | □ | □ |
| MKP | □ | □ | □ | □ | □ | □ | □ |
| HPP | □ | □ | □ | □ | □ | □ | □ |

Here we would like you to select what in your opinion is the most successful collaboration your company has had in the PGSEZ within the last 5 years and answer questions 5 to 16 accordingly (Table A3).

**Table A3.** Successful Collaborations.

5    How long did the collaboration last?

6    Is the collaboration still ongoing?
□ Yes                                        □ No

7    Which organizations were involved in the collaboration? (Multiple answers possible)

☐ Consulting company          ☐ Cluster management

☐ Province-level governmental organizations    ☐ National-level governmental organizations

☐ No other organization was involved    ☐ Others, please specify ………………………

8     Who decided to start the collaboration?

☐ Your company          ☐ The other company

☐ Consulting company          ☐ Cluster management

☐ Governmental organizations          ☐ Others, please specify …………………………

☐ I do not know

9     What was the manner of agreement for the collaboration?

☐ Formal contract          ☐ Informal mutual agreement

10    Who has monitored and assessed the outcomes of collaboration? (Multiple answers possible)

☐ Your company          ☐ The other party

☐ Consulting company          ☐ Cluster management

☐ Governmental organizations          ☐ Others, please specify ………………………

☐ The outcomes were not evaluated          ☐ I do not know

11    Was there any investment needed in infrastructure for the collaboration to take place?

☐ Yes          ☐ No

12    If yes, who was responsible for providing the required infrastructure (e.g. road transport road, pipeline) for the collaboration? (Multiple answers possible)

☐ Your company          ☐ The other party

☐ Consulting company          ☐ Cluster management

☐ Governmental organizations          ☐ Others, please specify ………………………

☐ I do not know

13    What was the method of communication during the collaboration? (Multiple answers possible)

☐ Formal meetings          ☐ Social media

☐ Shared database          ☐ Phone calls

☐ Written reports          ☐ Emails

☐ Others, please specify ………………………

How would you evaluate the quality of collaboration regarding:

14    shared strategic vision between the companies

☐ Has decreased or damaged     ☐Has not changed          ☐ Has significantly improved

15    long term relationship between the companies

☐ Has decreased or damaged     ☐Has not changed          ☐ Has significantly improved

16    information exchange platform between the companies

☐ Has decreased or damaged     ☐Has not changed          ☐ Has significantly improved

Please indicate to what extent each parameter would encourage your company to start new Industrial Symbiosis collaborations with existing or future companies in the PGSEZ (Table A4).

**Table A4.** Drivers of Industrial Symbiosis.

17　Increase in resource prices (energy, water, raw material)
　　☐Not at all　　　☐Slightly　　　☐Moderately　　　☐Very　　　☐Completely

18　Increase in waste disposal costs (e.g. landfill tax or carbon tax)
　　☐Not at all　　　☐Slightly　　　☐Moderately　　　☐Very　　　☐Completely

19　Resource scarcity (land, water, energy…)
　　☐Not at all　　　☐Slightly　　　☐Moderately　　　☐Very　　　☐Completely

20　New short or long term business opportunities for your company
　　☐Not at all　　　☐Slightly　　　☐Moderately　　　☐Very　　　☐Completely

21　Increasing energy efficiency or material productivity of your company
　　☐Not at all　　　☐Slightly　　　☐Moderately　　　☐Very　　　☐Completely

22　Information about available waste heat and material in the cluster for exchange
　　☐Not at all　　　☐Slightly　　　☐Moderately　　　☐Very　　　☐Completely

23　Decreasing Greenhouse Gas (GHG) emissions of the company
　　☐Not at all　　　☐Slightly　　　☐Moderately　　　☐Very　　　☐Completely

24　Readiness of required infrastructures (e.g. road transport road , pipeline) to carry out the exchanges
　　☐Not at all　　　☐Slightly　　　☐Moderately　　　☐Very　　　☐Completely

25　Information about other successful industrial symbiosis projects (national or worldwide)
　　☐Not at all　　　☐Slightly　　　☐Moderately　　　☐Very　　　☐Completely

26　Financial stimulation policies by the government (e.g., subsidies or tax cuts)
　　☐Not at all　　　☐Slightly　　　☐Moderately　　　☐Very　　　☐Completely

27　Monitoring and environmental assessment by governmental organizations (e.g. stack gas monitoring, waste water quality monitoring)
　　☐Not at all　　　☐Slightly　　　☐Moderately　　　☐Very　　　☐Completely

28　Cluster development plans organized by the cluster management
　　☐Not at all　　　☐Slightly　　　☐Moderately　　　☐Very　　　☐Completely

29　Workshops, conferences, seminars in the cluster to enhance networking and awareness of IS
　　☐Not at all　　　☐Slightly　　　☐Moderately　　　☐Very　　　☐Completely

　　Please mention below any additional issues related to implementation of Industrial Symbiosis in this cluster that you think are important and were not included in this questionnaire.

## Appendix B

**Table A5.** ADICO coding sample translated to English.

| Regulation. | Clause | Code | Description |
|---|---|---|---|
| Sixth country development plan | 38 | A | The government |
| | | D | must |
| | | I | Completion and implementation of wastewater and sewage collection, treatment, recycling, and management facilities in cities, industrial parks, service areas, and other units which generate swage with pollution level higher than national standards limit through contracting for sale or pre-sale of sewage discharge from existing facilities or future development plans. |
| | | C | --- |
| | | O | --- |
| | | Type | N |
| The executive procedure of waste management | 12 | A | Production units, using recycled raw material |
| | | D | --- |
| | | I | will be exempt from payment of determined charges. |
| | | C | for the use of such materials |
| | | O | --- |
| | | Type | S |
| Soil Protection law (Act) | 13 | A | Managers of Free Trade Zones, Industrial and Special Economic Zones, and Industrial parks |
| | | D | must |
| | | I | eliminate pollution and destruction within the scope of this Act and submit a report of actions to the DOE or ministry as appropriate. |
| | | C | In cases of pollution or destruction of soil is reported by DOE or the ministry |
| | | O | --- |
| | | Type | N |
| Energy consumption pattern reform law | 69 | A | The ministry of energy in collaboration with the ministry of industry, mine, and trade |
| | | D | must |
| | | I | plan and conduct practical training courses in general energy management and specific heat and electricity management for energy managers of industrial units in the national training center of energy management in industry, and grant a certificate to the trainees. |
| | | C | --- |
| | | O | --- |
| | | Type | N |
| Rules and regulations for the establishment of production, industrial and mining units | 12 | A | All Executive Organizations |
| | | D | --- |
| | | I | promote industrial plants to settlement in industrial areas and prevent the dispersal of these plants. |
| | | C | --- |
| | | O | --- |
| | | Type | S |

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
