# Peer review of "Institutional Lens upon Industrial Symbiosis Dynamics: The case of Persian Gulf Mining and Metal Industries Special Economic Zone"

_sustainability, doi:10.3390/su12156192_

Round 1
Reviewer 1 Report
This is a well-written article presenting a well-structured case study research on a topic of increasing importance. Important work is currently being done setting up the stage for study of emergence and dynamics (restructuring) of industrial symbiotic networks as socio-technical phenomena. The development is done in 1. new conceptualisations, 2. in research methods and 3. is developing in direction of development of tools for science-informed policy-making for circular economy. This may make the article visible in the scientific community.
However, the article does not sufficiently mention the extent of these significant developments and existence of other possible approaches and as a result underestimates the importance of its own contribution. This is to some extent understandable since this is a case study. In order to improve it, it would be good to point to other conceptualisations and methodological approaches, which also allow comparative research. One such case is present in the following two works, 1. drawing on social fields theory, extending analysis of social aspects to institutions, social networks and cognitive frames, 2. using non-social aspects in the analysis and 3. applying new tools for analysis of IS networks and informing policy.
1. Mileva Boshkoska, Biljana; Rončević, Borut; and Džajić Uršič, Erika (2018): Modeling and evaluation of the possibilities of forming a regional industrial symbiosis networks. Social sciences, ISSN 2076-0760, 2018, vol. 7, iss. 1. https://www.mdpi.com/2076-0760/7/1/13/pdf, doi: 10.3390/socsci7010013.
2. Džajić Uršič, Erika (2020): Morphogenesis of Industrial Symbiotic Networks. Berlin: Peter Lang.
Other approaches may also be mentioned in the paper.
Author Response
Dear Reviewer,
Thanks for providing constructive comments and relevant references. While many previous studies investigated existing IS networks, this research focuses on emerging industrial clusters (EICs) in which IS is not shaped yet. This paper aims to get insight into probable IS dynamics in an EIC through investigating social forces shaping IS collaborations. It offers a detailed and systematic approach to study actors' motives, the structure of previous collaborations, and institutions. It shows how linking the IS dynamics framework to ADICO grammar of institutions improves our understanding of IS emergence circumstances. The scientific contribution of the paper is elaborated by providing more references and clarifying the novelty of the research. (see lines 61-63, 65-66, and 85-88)
Reviewer 2 Report
The document shows interesting research ideas although it is very advanced in the documentation of the data and above all in the investigations on the area of interest, both from a scientific, political, economic and financial point of view.
The environmental point of view is a fundamental economic driver for the region, due to the scarcity of resources, the IS strategy is successful if well-articulated and above all, if shared among the stakeholders. Many social and cultural obstacles slow down the advance of IS. The research will focus on the definition of socio-technical measures for the implementation of IS in the region.
Author Response
Dear Reviewer,
Thanks for your positive feedback.